# Constructing a knowledge-driven and data-driven hybrid decision model for etiological diagnosis of Ventricular Tachycardia

Min Wang[1], Zhao Hu[2], Xiaowei Xu[1], Si Zheng[1], Yan Yao[2*], Jiao Li[1*]

1 Institute of Medical Information/Library, Chinese Academy of Medical Sciences & Peking Union Medical College, Beijing 100020, China.

2 Fuwai Hospital, Chinese Academy of Medical Sciences & Peking Union Medical College/National Center for Cardiovascular Diseases, Beijing, China.

**Correspondence information:**

**Yan Yao, Ph.D.**

Fuwai Hospital, Chinese Academy of Medical Sciences & Peking Union Medical College

Email: ianyao@263.net.cn

Phone: 010-88322401

**Jiao Li, Ph.D.**

Institute of Medical Information/Library, Chinese Academy of Medical Sciences & Peking Union Medical College

Email: li.jiao@imicams.ac.cn

Phone: (+86)18618461596

# Abstract

**Background:** Each of the two clinical decision support models, data-driven and knowledge-driven, has its own unique strengths and challenges, and at the same time, the development of People-Centric Artificial Intelligent (PCAI) clinical decision support urgently needs to explore the effective integration strategies of the two driven models.

**Objective:** Constructing a trustworthy and highly accurate hybrid decision model incorporating knowledge-driven and data-driven model, and applying it to the field of healthcare.

**Methods:** We collected authoritative clinical practice guidelines, expert consensus and medical literature in the field of cardiovascular diseases as knowledge sources and retrospectively collected electronic medical record information of patients with ventricular tachycardia (VT) from Fu Wai Hospital as a dataset. The knowledge-driven model constructs a clinical pathway using a knowledge rule-based approach, and the data-driven model constructs a multi-classification machine learning model for etiological diagnosis of VT based on real-world data. The hybrid model's uses the clinical pathway as the basic framework, and the machine learning model is embedded as a custom operator into the decision node of the process. The comparison metrics of the three models are precision, recall and F1 score.

**Results:** In this study, we selected three clinical guidelines as the knowledge source for the knowledge-driven models, as well as collected 1,305 patient data as the dataset. A total of five machine learning models were constructed and the best model was XGBoost model (precision, recall, and F1 were 88.4%, 88.5%, and 88.4%, respectively. The hybrid model adopts the knowledge-driven thinking, embedding the machine learning model into the decision-making node of the two layers of classification, respectively. The precision, recall and F1-scores for the knowledge-driven model were 80.4%, 79.1% and 79.7%; for machine learning model were 88.4%, 88.5%, and 88.4%; for hybrid model were 90.4%, 90.2% and 90.3%.

**Conclusion:** The results show that the strategy of integrating knowledge-driven and data-driven clinical decision-making models is feasible. Compared to the pure knowledge-driven and data-driven models, the hybrid model demonstrated higher accuracy, and all the decision-making results of the model were based on evidence-based evidence, which was closer to the actual diagnostic thinking of clinicians. This new generation of PCAI systems for clinical decision-making needs to be applied to a wider range of medical fields and rigorously validated in the future.

**Keyword** Ventricular Tachycardia, knowledge-driven, data-driven, machine learning, hybrid model, decision-making

# Introduction

With the development of medical science and information technology, clinical decision-making plays a key role in medical practice. Clinical decision-making refers to a series of complex processes, such as disease analysis, diagnostic inference, and treatment plan development, when clinicians are faced with a patient's condition, based on existing medical knowledge, the patient's specific situation, and the available medical resources[1]. This process not only covers medical theories and statistical principles, but may also incorporate multidisciplinary knowledge and

technological methods such as Artificial Intelligence (AI), so as to construct a set of scientific and rigorous clinical decision-making system. Clinical Decision Support System (CDSS), as an important tool to assist clinical decision-making, plays a significant role in optimizing the diagnosis and treatment process and enhancing the accuracy of decision-making. Currently, mainstream CDSS are categorized into knowledge-driven and data-driven types mainly based on their operational mechanisms[2].

It can be seen that current models for both knowledge-driven and data-driven decision making fail to achieve the desired goals due to technical challenges (e.g., lack of decision accuracy),low quality of evidence-based evidence, and system design failures (e.g., unfriendly human-computer interaction design)[3]. These challenges lead to illogical and inappropriate decision making by clinicians, which increases uncertainty, decreases the quality of decisions[4], and affects job satisfaction with decision support systems. In healthcare, knowledge-based driven systems are more acceptable to clinicians, while data-based driven systems are more accurate. In 2019, the International Organization for Standardization (ISO) advocates the adoption of a human-centered design approach in the development of interactive systems, such as AI-based clinical decision support systems. The approach aims to make systems both easy to use and practical by focusing on the user needs and requirements, applying knowledge and techniques from the fields of human factors and usability[5]. People-Centric Artificial Intelligent (PCAI) as an emerging concept has demonstrated its core value in the field of clinical decision support. PCAI is dedicated to the development of AI systems that augment, rather than replace, human capabilities, and to ensure that AI technology always assists the judgment of healthcare professionals in clinical decision support processes that are transparent and fairly operation, and with full respect for patient privacy[6]. The ISO standard describes key principles of human-centered design that aim to enhance the utility, ease of use, and utilization of relevant healthcare technologies to improve health outcomes and impact. Technologies should be designed with the user in mind who can, wants or needs to use the technology, rather than requiring the user to significantly change behaviors or attitudes to accommodate the technology[7].

The two types of clinical decision support models, data-driven and knowledge-driven, each have their own strengths and challenges, and the U.S. Department of Defense Advanced Research Projects Agency (DARPA) proposed a pathway for integrating the two strategies, knowledge-based and data-based, in 2017. This initiative aims to develop a dual knowledge- and data-driven strategy model by combining the rigorous logic rules of traditional knowledge engineering with the powerful learning capabilities of data-driven approaches to build more transparent and efficient AI systems. The development of digital clinical decision support needs to seek an effective fusion of the two driving models to build trusted, accurate, and personalized next-generation human-centered CDSS to meet the growing demand for intelligent clinical decision aids in the development of healthcare[8, 9].

The application of clinical decision support in the field of arrhythmia is very promising. According to the World Health Organization (WHO), about 17 million people die each year due to cardiovascular diseases[10], accounting for about one-third of all global deaths. Early symptoms of cardiovascular diseases are mostly arrhythmias, also known as arrhythmias[11], and the diagnosis of arrhythmias aims to improve symptoms, quality of life (QOL), and prognosis by preventing sudden cardiac death due to fatal ventricular arrhythmias. Among these, ventricular tachycardia (VT) is a common electrocardiographic manifestation of ventricular disease and a type of arrhythmia[12]. The

etiological diagnosis of VT involves multiple aspects, including patient history, signs, electrocardiogram (ECG), imaging, and even biopsy of the myocardium and other tissues[13] and the variety of underlying diseases and clinical manifestations of VT, which are prone to hemodynamic instability, make the etiological diagnosis of VT a major challenge in clinical decision making[14-16].

Based on the above, the two clinical decision support models, data-driven and knowledge-driven, each have their own unique advantages and challenges as well as the need for a hybrid driven model, and the development of human-centered clinical decision support urgently needs to explore the way to effectively integrate the two driven models. The goal of this paper is to construct a trustworthy and highly accurate hybrid decision-making model that integrates both knowledge- and data-driven models, and to apply it in the field of arrhythmia diagnosis and treatment.

# Method

## User Requirements Analysis

The clinical scenario addressed in this study is the etiological diagnosis of VT in the context of arrhythmia diagnosis and treatment. The diagnosis of VT is extremely challenging, and CDSS have emerged as an ideal tool to enhance the diagnostic capabilities of cardiologists. In our previous work, we completed a requirements survey on the knowledge and practice in VT diagnosis of 687 cardiologists in China[17]. A total of 567 valid responses were analyzed. Chinese cardiologists had significant deficiencies in VT knowledge and practice, and the knowledge assessment showed that 383 cardiologists (68%) lacked knowledge in diagnostic assessment, and had an urgent need for digital decision-making tools, with the majority of cardiologists (60.7%) indicating a need for assistive tools such as CDSS.

## Knowledge-driven model

We have proposed a digital clinical guideline representation tool in our previous work[18]. The tool mainly defines data acquisition nodes, decision nodes, action nodes, composite nodes, and interpretation nodes. The core steps are as follows: (1)Knowledge acquisition. Knowledge source inclusion criteria: Applicability to clinical scenarios, within defined knowledge domains and subdomains, publishing source document entities, actionable clinical recommendations, clinical experts' views on the relevance, potential impact, and scope of the recommendations to prioritize the selected recommendations. (2)Knowledge extraction. Identification of the knowledge of the textual format guideline and extraction of key clinical concepts are accomplished by using natural language processing techniques and a large language model interface. (3)Mapping process. Different types of knowledge are represented using different shapes of nodes defined by the digital clinical guideline representation tool. Based on the extracted and represented knowledge, explicit rules or conditions are developed for each decision node. The rules in the clinical pathway are written through if-then structures and they are connected through Boolean logic operators. Finally the different types of node graphs and logical relationships are visualized into an easy-to-understand clinical pathway diagram, including all key decision points and possible branches. (4) Rule Binding. The rules are bound to the corresponding clinical pathway to achieve a computer-executable clinical pathway model.

In the clinical scenarios of this study, we included a total of three guidelines as a source of

knowledge[13, 14, 19], and where necessary, supplemented the information using other sources of data (e.g., systematic reviews and meta-analyses) to strengthen the recommendations in certain aspects not fully covered by the guidelines. This research team's panel of cardiologists has mapped a prototype clinical pathways based on the knowledge source and constructed a computer-executable model using a digital clinical guideline representation tool.

## Data-driven model

### Data collection and preprocessing

The retrospective data that was used for training the machine learning algorithms was provided by the Fuwai Hospital, CAMS, China. Patients with ventricular tachycardia at the Arrhythmia Centre of Fuwai Hospital were consecutively admitted between January 1, 2013, and September 1, 2023. The inclusion criteria were as followed: patients with a discharge diagnosis that included "ventricular tachycardia". The patients' records comprised their clinical data, including medical history, vital signs, current medications, electrocardiograms (ECGs), echocardiograms, and laboratory test results, all of which were diagnosed by professional physicians. The VT due to ischemia heart diseases were defined as whose diagnosis containing "myocardial infarction" or "myocardial ischemia". The VT due to non-ischemic structural heart diseases contained "myocarditis", "cardiac amyloidosis", "cardiac sarcoidosis", "non-compaction cardiomyopathy", "cardiomyopathy". The idiopathic VT is defined according to the discharge diagnosis as "idiopathic ventricular tachycardia". All the etiological diagnosis label was reviewed by the cardiologists, which regarding as the golden standard.

### Machine Learning Model Construction

In order to develop a model for etiological diagnosis of VT, we applied 5 different ML models for supervised learning (implementation in the Python Scikit-Learn): (1) Logistic regression. (2) Random forest[20]. (3) XGBoost[21]. (4) Light Gradient Boosting Machine (LightGBM)[22]. (5) Support Vector Machine (SVM)[23]. The model was constructed using GridSearchCV to optimize the hyperparameters.

Considering the metrics for VT etiological diagnosis, we will use a weighted macro-average approach to calculate assessment metrics to solve the triple classification problem. We will use the precision, recall and F1-score derived from the ten-fold cross-validation as model evaluation criteria.

## Hybrid model

The hybrid model takes the clinical pathways as the basic framework, integrates the advantages of knowledge-driven and data-driven, tries to embed the machine learning model into the decision-making nodes of the pathways, and increases the remark statements in the corresponding nodes to use the corresponding custom operators, and finally forms the hybrid model. Specific construction steps include: (1) Determine the integration strategy. Specify the strategy for integrating knowledge and data, and decide when and how to incorporate machine learning models in the knowledge-driven pathways. (2) Integrate the knowledge-driven model to ensure that the clinical guidelines, expert consensus and pathways have been sufficiently structured and modularized to facilitate interfacing with the machine learning model. Each decision node should have its inputs and outputs and trigger conditions clearly defined. (3) Design machine learning model decision nodes. Selecting appropriate

decision nodes so that the machine learning model can be seamlessly embedded into the knowledge-driven model as custom operators. (4) Embedding machine learning models. Machine learning models are integrated at key decision nodes and triggered to run based on predefined rules and conditions. The necessary condition for triggering is that the machine learning model can be triggered only when the rules on the decision node need to be in line with the knowledge model (based on evidence-based evidence) and the existing rules are not able to make an accurate decision. (5) Result output. At the decision point, combining the knowledge-driven rule judgement with the prediction result of the machine learning model, the priority of the decision is based on the knowledge-driven model result first, and then on the machine learning model result. (6) Validation and optimization. Validate the hybrid model using an independent patient dataset to evaluate its performance metrics such as accuracy, recall, F1 score, etc. Adjust the fusion strategy, machine learning parameters, or knowledge rules based on the feedback for iterative optimization.

In this study, the hybrid model uses the arrhythmia diagnosis model as the base framework, and the machine learning model for etiological diagnosis of VT is embedded as a custom operator in the decision node of the arrhythmia diagnosis process. The machine learning model on its own uses a direct 3 classification (ischemia heart diseases, non-ischemic structural heart diseases, and idiopathic VT) for etiological diagnosis of VT. In the knowledge-driven clinical pathways, given the multifaceted origins of VT can be systematically classified into ischemic heart diseases, non-ischemic structural heart diseases, and idiopathic VT[13, 24-26]。 The first two categories fall under the umbrella of structural heart pathologies, in contrast, idiopathic VT characterizes those cases where VT arises in the absence of structural heart disease [27]. There are two differential diagnosis strategies commonly used in clinical practice which can guide the subsequent management, i.e., ischemia and non-ischemia, and idiopathic versus non-idiopathic (structural), which is also in accordance with the actual clinician's thinking. Therefore, the hybrid model adopts the knowledge-driven thinking when integrating the knowledge and data-driven model, according to the classification of the disease into ischemia and non-ischemia, and then into idiopathic and non-idiopathic (structural) for the second layer of classification, embedding the machine learning model into the decision-making node of the two layers of classification, respectively, and ultimately classified into ischemia heart diseases, non-ischemic structural heart diseases, and idiopathic VT.

## Statistical analysis

Statistical analyses were performed using Python (version 3.9), and the differences were considered statistically significant at P<0.05. Continuous variables such as age, BMI and other measures that conformed to normal distribution were expressed as means, and the t-test was used for comparison between groups; count data of categorical variables such as sex and comorbidities were expressed as frequencies (%), and the $\chi^2$ test was used for comparison between groups. To rigorously compare the performance of the three models and to validate the efficacy of the hybrid model, the dataset (n=1305) was randomly partitioned into test dataset (n=783) and validation dataset (n=522). Subsequently, stratified by the etiological diagnostic outcomes of VT, the precision, recall, and F1 scores were evaluated.

# Results

## Clinical Pathway

Figure 1 shows the knowledge-driven clinical pathway prototype. In order to effectively assess the validity of this knowledge model and to compare it with other models, we evaluated the knowledge-driven model against the available data and the performance results are shown in Table 2.

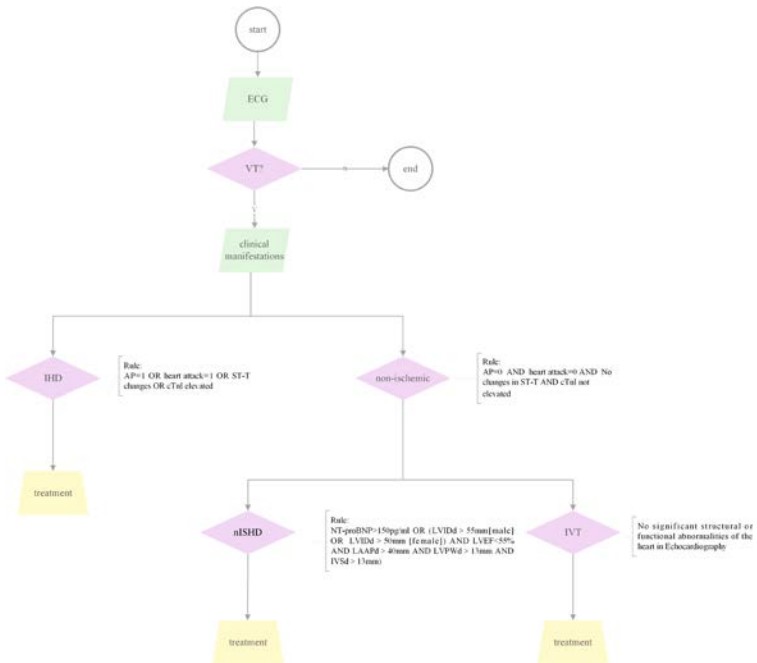

Figure 1 Knowledge-driven clinical pathway

Note: ECG: electrocardiography; VT: ventricular tachycardia; IHD: ischemic heart diseases; nISHD: non-ischemic structural heart diseases; IVT: idiopathic ventricular tachycardia;

## Machine learning model

The performance of the VT etiological diagnostic machine learning model is shown in Table 1. The XGBoost model achieved better performance in terms of evaluation metrics precision, recall, and $F1$ on the three ventricular tachycardia etiological diagnostic works of interest, with $F1$ reaching 89.7%, 76.9%, and 94.1% of the experimental performance. From the results of the overall performance comparison of the model, the XGBoost model also achieved the best performance (precision, recall and F1 scores of 88.4%, 88.5%, 88.4% respectively).

Table 1 Comparison of model performance

| Model | Ischemic heart diseases | | | Non-ischemic structural heart diseases | | | Idiopathic VT | | | Overall | | |
|---|---|---|---|---|---|---|---|---|---|---|---|---|
| | P | R | F1 | P | R | F1 | P | R | F1 | P | R | F1 |
| Logistic Regression | 0.735 | 0.800 | 0.766 | 0.562 | 0.360 | 0.439 | 0.764 | 0.824 | 0.824 | 0.709 | 0.722 | 0.710 |
| Random Forest | 0.884 | 0.884 | 0.884 | 0.746 | 0.707 | 0.726 | 0.934 | **0.971** | **0.952** | 0.871 | 0.872 | 0.871 |
| XGBoost | **0.879** | **0.916** | **0.897** | 0.809 | **0.733** | **0.769** | **0.950** | 0.931 | 0.941 | **0.884** | **0.885** | **0.884** |

| | | | | | | | | | | | | |
|---|---|---|---|---|---|---|---|---|---|---|---|---|
| LightGBM | 0.879 | 0.912 | 0.895 | **0.815** | 0.707 | 0.757 | 0.933 | 0.951 | 0.942 | 0.881 | 0.883 | 0.881 |
| SVM | 0.840 | 0.879 | 0.859 | 0.823 | 0.68 | 0.745 | 0.848 | 0.873 | 0.860 | 0.839 | 0.839 | 0.837 |

## Hybrid model

The fusion of knowledge-driven and data-driven clinical pathway is shown in Figure 1. For better understanding, we will explain the hybrid model inference process: when facing a new patient with VT, the inference is carried out step by step in accordance with the knowledge-driven pathway, and when the pathway proceeds to the critical decision node, the system firstly makes a decision based on the knowledge-driven rules, and if the decision rule fails to cover the patient's information, then the machine learning model is triggered to make a decision. At least a knowledge-driven (or knowledge and data-driven) result label based on the knowledge is output at the decision node at that stage, and then the rule path is continued from the branch corresponding to that label until the final node, completing diagnosis and treatment. The output of the final hybrid model is a fusion of both knowledge- and data-driven model decision results and the corresponding treatment plan as the system's recommended treatment plan.

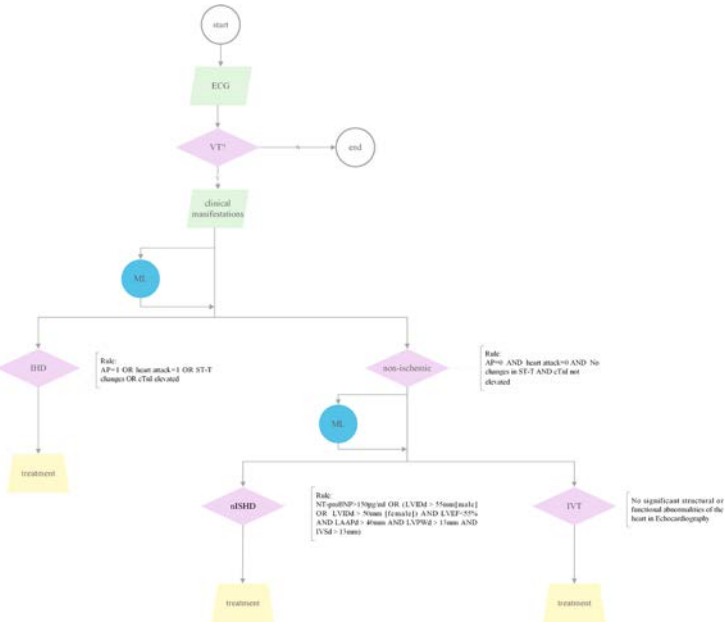

Figure 2 Clinical pathways based on hybrid model

Note: ECG: electrocardiography; VT: ventricular tachycardia; IHD: ischemic heart diseases; nISHD: non-ischemic structural heart diseases; IVT: idiopathic ventricular tachycardia; ML: machine learning;

Table 2 shows the performance comparison results of the 3 models, and we choose XGBoost as the data-driven model's result for comparison. The results show that the hybrid model is better than the other two models in terms of precision, recall and F1 score.

Table 2 Performance comparison of three models

| Model | P | R | F1 |
|---|---|---|---|
| Knowledge-driven model | 0.804 | 0.791 | 0.797 |
| Data-driven model (ML) | 0.884 | 0.885 | 0.884 |

| | | | |
|---|---|---|---|
| **Hybrid model** | **0.904** | **0.902** | **0.903** |

# Discussion

In this study, we constructed a trustworthy and highly accurate knowledge- and data-driven hybrid decision-making model using knowledge-based clinical pathways as a basic framework, and then embedded machine learning models.

In view of the respective strengths and limitations of knowledge-driven and data-driven models, the combination of the two is considered complementary, and synergies between rule-based systems and machine learning have been demonstrated[28]. In recent years, in the field of cardiovascular disease, researchers have also begun to explore combined knowledge- and data-driven decision support methods. A Korean team has published research in recent years on a CDSS for Cardiovascular Disease, which used a hybrid (expert-driven and machine-learning-driven) knowledge acquisition approach to build a knowledge base and demonstrated the potential of this hybrid model in assisting clinicians with heart failure diagnosis[29].

All clinical diagnosis and treatment processes of the hybrid model constructed in this study are based on a knowledge-driven model, and the machine learning model is embedded as a custom operator in the decision-making nodes along clinical pathways, which is used to supplement the knowledge-driven model that is difficult to formalize or quantify. The advantage of PCAI decision making with hybrid models is reflected in the fact that the fusion strategy used means that it ensures that all decision-making actions are based on evidence-based evidence, and guarantees a certain degree of decision-making accuracy. At the same time, this kind of decision-making thinking is closer to the real process of clinicians' rational (guideline-compliant) and perceptual (clinical experience) co-decision-making. In addition, the model can be applied to more practical problems to enhance the usability of the CDSS system. It is worth noting, however, that regardless of whether the model is knowledge-driven, data-driven, or hybrid-driven, the system needs to clearly communicate the decision-making situation to the clinician, who will make the final clinical decision.

# Conclusion

In this study, knowledge-driven and data-driven clinical decision-making models were effectively fused to construct a trustworthy and highly accurate hybrid model, and the accuracy of its decision-making results were compared with those of knowledge rules and machine learning models. The results show that this fusion strategy is feasible and the hybrid model exhibits higher accuracy compared to the rule-only and machine learning-only models, and all the decision results of the model are based on evidence-based evidence, which is closer to the actual diagnostic thinking of clinicians. This new generation of PCAI systems for clinical decision-making can not only improve the quality and efficiency of healthcare services, but also provide clinicians with trusted decision support, which ultimately leads to personalized treatment services for patients, and thus promotes the modernization and intelligence of the healthcare industry. In the future, this integration strategy needs to be applied to a wider range of medical fields and rigorously validated.

# Funding

This work was supported by the National High Level Hospital Clinical Research Funding[grant numbers 2022-GSP-GG-25]; and the CAMS Innovation Fund for Medical Sciences (CIFMS) [grant number 2021-I2M-1-056].

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
