# OpenReview forum: "Constructing a knowledge-driven and data-driven hybrid decision model for etiological diagnosis of Ventricular Tachycardia"
_KDD.org/2024/Workshop/AIDSH — KDD-AIDSH 2024 Poster_

### Official Review · Reviewer_DuPk · 2024-06-09
**Comments for submission 12 in AIDSH**

**Rating:** 8
**Confidence:** 5

**Review:**

Authors combine the knowledge-driven and data-driven method in VT detection. I am very interested in this research, but there are some problems limiting the manuscript quality.\
1.The language expression should be clear enough. For example, in the results located in the abstract, this sentence could be deleted,"(precision,recall, and F1 were 88.4%, 88.5%, and 88.4%, respectively".\
2.The resolution of figures are so low that I cannot get any useful information. If possible, authors should provide a high-resolution version, which could demonstrate the knowledge-driven clinical pathway.\
3.I am interested in the knowledge-driven pathway,  could you provide a more detailed explaination. If the ECG is given, how can I judge it as VT or other diseases. Because the figure 1 and figure 2 is is really blurry, I can't even see the text clearly.\
4.The fusion process of hybrid model needs to be clearly demonstrated. Are they quantitative or qualitative fusion? Is there a weighted synergy between the two pathway.\
5.Could you show the experimental results about the hybrid model with other machine learning models? It may prove the advantages of the proposed method from many ascepts.\
6.It is excepted to compare with other methods in this research field.\
7.In addition, the hyperparameter configuration is lost in the current version, though authors mentioned "The model was constructed using GridSearchCV to optimize the hyperparameters". Then, how about the range of hyperparameter values?

---

### Official Review · Reviewer_fyve · 2024-06-18
**Combining logical rules and machine learning models for VT diagnosis, but the proposed method lacks innovation**

**Rating:** 6
**Confidence:** 4

**Review:**

This work focuses on diagnosing a special arrhythmia in the field of heart disease, ventricular tachycardia (VT), which is a representative and important problem. The proposed method combining expert knowledge with machine learning is also a hot research issue that has not been well resolved. This work mainly establishes a series of diagnostic rules into a clinical pathway for VT diagnosis based on expert knowledge. Then it uses the machine learning model as a supplement to the rule discrimination. When the rules are inconsistent with the data, the machine learning model determines some local conclusions.

There are some shortcomings:

1. The article is generally readable, but some parts are not very clear.
- Such as "mostly arrhythmias, also known as arrhythmias" in the 3rd paragraph of the "Introduction" section;
- In the 2nd paragraph of the "Introduction" section, a lot of space is used to describe ISO standards and PCAI concepts, which is not quite relevant to this work;
- The fonts of Figure 1 and Figure 2 are too small to see the content of the rules;
- The 1st sentence of the 2nd paragraph in SubSection "Knowledge-driven Model" of the Section "Method" is long and difficult to read, and it needs further polishments.

2. The article says "the rules on the decision node need to be in line with the knowledge model (based on evidence-based evidence) and the existing rules are not able to make an accurate decision" However, how to activate the machine learning model for supplementary judgment is not explained clearly.
In the experiment, the data included medical history, vital signs, current medications, electrocardiograms (ECGs), echocardiograms, and laboratory test results. How to process and utilize these heterogeneous data was not clearly expressed.

3. The way of combining the machine learning model with the logical rules in this work is a bit trivial and lacks innovation. In addition, the hybrid model proposed is only for VT disease diagnosis, and it is difficult to adapt to the diagnosis of other diseases.

4. This work collected 10 years of patients with a discharge diagnosis that included "ventricular tachycardia" from Fuwai Hospital. The data volume is only 1305 samples, of which the validation data set size is only 522. How to ensure that there is no overfitting? It is recommended to do n-fold cross-validation and supplement experiments with larger public data sets. VT-related patients can be extracted from public data sets.

5. The article begins by stating that building a hybrid model is both trustworthy and highly accurate, but the experiment only verifies that the fusion model has a certain improvement in the accuracy of the results, but does not discuss whether the model's credibility or interpretability of the original clinical pathway method will be lost after adding the machine learning model.

---

### Decision · Program_Chairs · 2024-06-28

Accept (Poster)